# Trends of Perinatal Stress, Anxiety, and Depression and Their Prediction on Postpartum Depression

**DOI:** 10.3390/ijerph18179307

**Published:** 2021-09-03

**Authors:** Ching-Yu Cheng, Yu-Hua Chou, Chia-Hao Chang, Shwu-Ru Liou

**Affiliations:** 1College of Nursing, Chang Gung University of Science and Technology, Puzi City 61363, Taiwan; cycheng@mail.cgust.edu.tw (C.-Y.C.); chchang@mail.cgust.edu.tw (C.-H.C.); 2Chiayi Chang Gung Memorial Hospital, Puzi City 61363, Taiwan; 3Department of Nursing, Cardinal Tien Junior College of Healthcare and Management, Yilan County 26646, Taiwan; chouyuhua57@gmail.com

**Keywords:** perinatal, stress, anxiety, depression, pregnancy, postpartum

## Abstract

Perinatal stress, anxiety, and depression impacts not only women but also their child(ren). The purpose of this longitudinal study is to explore trends of stress, anxiety, and depressive symptoms from pregnancy to postpartum and understand predictions of stress and anxiety on postpartum depression. One-hundred-fifty-six women at 23–28 weeks gestation (T1), 147 at 32–36 weeks gestation (T2), 129 at over 36 weeks gestation (T3), and 83 at postpartum (T4) completed study surveys. The Perceived Stress Scale, Center for Epidemiologic Studies Depression scale, and State-Trait Anxiety Inventory were used to measure stress, depressive symptoms, and anxiety. Descriptive statistics, Pearson and Spearman’s correlation, and Generalized Estimating Equation were applied to analyze the data. Results showed that levels of anxiety and depressive symptoms increased from 24 weeks gestation to postpartum, whereas stress levels decreased during pregnancy but increased in postpartum. Over half of women experienced anxiety symptoms, especially during late pregnancy and postpartum. Stress, anxiety, and depressive symptoms were inter-correlated. Notably, women at late pregnancy and postpartum were prone to stress, anxiety, and depression. Prenatal anxiety could predict postpartum depressive symptoms. Active assessment and management of stress, anxiety, and depression is needed and should begin from early pregnancy and continue until postpartum.

## 1. Introduction

Based on a 2017 worldwide statistic, depressive disorders were among the top three leading cause of years lived with disability [1]. Compared with men, women have a higher prevalence and incidence of depression [2,3,4]. Specifically, during the perinatal period, women had a higher risk of developing or aggravating an underlying mental illness [5]. Systematic review and meta-analysis articles reported that pooled prevalence of depression was 7.4%, 12.8%, and 12.0% at the first, second, and third trimester of pregnancy, respectively, and it was 17.7% at postpartum [6,7]. Yet, perinatal depression may not be recognized because of physical and psychological changes throughout pregnancy and postpartum [8]. Furthermore, only one-fourth of women who experienced moderate-severe postpartum depressive symptoms sought mental health care or consultations about their emotional or psychological problems [9]. Anxiety is another important perinatal mood disorder that needs to addressed [8] and is comorbid with depression [2,10]. A meta-analysis article reported that the prevalence of self-reported anxiety symptoms was 18.2%, 19.1%, and 24.6% at the first, second, and third trimester, respectively, while it was 17.8% during the first postpartum month [11]. 

Both perinatal depression and anxiety can lead to adverse outcomes. Prenatal depressed moods were related to a lower intensity of preoccupation with and more negative affective experiences toward the fetus [12], poor self-esteem during pregnancy [13], shorter gestation [14], birth complications with the baby and low birth weight [14,15,16]. Children of mothers with postpartum depression were at risk of maltreatment and subsequent social, emotional, and behavioral problems [17]. In particular, mothers who experienced depressed moods during the first trimester or throughout pregnancy had a higher level of postpartum psychosocial difficulties than their counterparts [18], and those who had postpartum depression had infant attachment problems [19]. Regarding anxiety, a higher level of prenatal anxiety is related to a shorter pregnancy duration [20]. Pregnant women who experienced anxiety had a higher risk of having a baby with birth complications and lower Apgar scores [14]. Mothers with anxiety were less likely to breastfeed and were less likely to breastfeed for longer than 1 month [21]. Perinatal anxiety was correlated with children’s social-emotional development, negative behaviors and emotions throughout the period from infancy to adolescence [19,22]. Women with untreated prenatal mood and anxiety disorders had a higher risk of suicide, preeclampsia, preterm birth, cesarean delivery, not breastfeeding or breastfeeding suboptimally [23], infant death, child behavioral and developmental disorders, obesity, asthma, and injuries [23]. However, Rees et al.’s systemic review on 14 articles found that the impacts of perinatal anxiety on the child’s emotional problems were low [24]. Although maternal anxiety or depression alone has a certain degree of influence, researchers have suggested that pregnant women with comorbid depression and anxiety have a greater incidence of prematurity than the only depressed or only anxiety groups [25]. The average cost per woman with untreated mood and anxiety disorders during pregnancy and five-year postpartum in 2017 was about US $32,300. Of this cost, over 50% occurred during pregnancy [23].

Studies have been done to explore factors related to perinatal emotional distress. Among the factors, perceived stress has been found to be related to depressive symptoms and anxiety from early pregnancy to postpartum [26,27]. This correlation exists not only in adults but also in late adolescents and young adults [28]. Specifically, postnatal stress predominantly explains the variance of postnatal depressive symptoms, anxiety, and parental self-efficacy [27]. Mothers who perceived higher levels of stress were at higher risk of having pregnancy complications [29,30] and preterm birth [30], were less likely to breastfeed [31], and increase disease risk of their offspring within first 7 months of life [32]. Postpartum mothers who experienced higher levels of stress reported less parental self-efficacy [27].

By understanding the trends and relationships of stress, depressive symptoms, and anxiety throughout the perinatal period, appropriate interventions can be implemented to prevent the consequent impacts of those mental distress on women and their offspring. The purpose of this study is to explore trends and relationships between perinatal depressive symptoms, anxiety, and stress and predictors of postpartum depressive symptoms. The following research questions are answered: (a) what are the trends of stress, depressive symptoms, and anxiety from pregnancy to postpartum? (b) what are the factors of postpartum depressive symptoms?

## 2. Materials and Methods

### 2.1. Design

This study is part of a prospective longitudinal study that focuses on perinatal mental distress, sleep quality, fatigue, and biomarkers. This paper reports perinatal stress, depressive symptoms, and anxiety from pregnancy to postpartum. The data were collected from May 2012 to September 2013. The participants were recruited when they were about 24 gestational weeks (T1) and were followed up at around 32 weeks gestation (T2), 36 weeks (T3), and one month postpartum (T4), with a total of four times surveyed. Because pregnant women may experience more physical symptoms during the third trimester when compared with the second trimester and the prevalence of depression remained high from the second to the third trimester whereas the prevalence of anxiety increased in the third trimester, the participants were surveyed one time in the second trimester (before 28 weeks gestation) and two times in the third trimester (after 28 weeks gestation). The time for postpartum survey was done after the participants were discharged from the hospital when they had to face the childcare responsibility and adjust to the new maternal role. Since new mothers would have their postnatal checkup at around 4 to 6 weeks postpartum, the survey appointment was made at that time point.

### 2.2. Setting

Participants were recruited at the obstetric clinic of two hospitals located in two metropolitan cities in northern (Taipei) and southern (Chiayi) Taiwan.

### 2.3. Sample and Sampling

Non-probability sampling was used to recruit participants. Pregnant women who were at least 18 years old, could read and write Chinese, were at about 24 weeks of gestation, were singleton, and did not have any pregnancy complications (including a diagnosis of prenatal depression or anxiety disorder) were eligible for participating in this study. The sample size was calculated using G*Power 3.0.10 [33]. To achieve a median effect size (.13) [34], power of 0.80, α of 0.05, one group, and median correlation among repeated measures (r = 0.3), 117 pregnant women were needed. Using a longitudinal design, this study might have high attrition rate of 30%; therefore, we needed to recruit at least 153 pregnant women. 

In total, we invited 160 pregnant women who were eligible to participate, 156 of whom participated and completed the T1 survey. We lost contact with 9, 18, and 46 participants at T2, T3, and T4, respectively; therefore, 147, 129, and 83 participants remained in the study at T2, T3, and T4, respectively. Those who dropped out and remained in the study did not differ in age, parity, marital status, educational level, employment, happiness about the pregnancy, whether they planned the pregnancy, and whether they had sleep disturbances; however, they differed in gestational age at T1. Those who dropped out from the study (M = 24.84, SD = 1.13) were more advanced in their pregnancy than those who participated throughout the study (M = 24.49, SD = 0.74, t = 2.20, *p* = 0.03). The attrition rate from T1 to T4 was 46.8%, which was higher than our expectation. We did a post-hoc power analysis based on the lowest correlation value between measured variables (r = 0.35 for both T1 and T4 trait anxiety, and T1 state anxiety and T4 stress) and sample size of 83, the power was 0.91, which meant an adequate sample size.

The mean age of the participants was 31.46 (SD = 4.24) years, and the mean gestational age at T1 was 24.63 (SD = 1.00) weeks. As shown in Table 1, around half of the participants were primiparous and had an educational level of bachelor’s degree or higher. Most of the participants were married, employed, happy about the pregnancy, had planned the pregnancy, and experienced sleep disturbances (especially interrupted sleep).

### 2.4. Instruments

Depressive symptoms. The 20-item Center for Epidemiologic Studies Depression (CESD) is a four-point (0–3) self-administered scale [35] and is used to measure depressive symptoms from prenatal to postpartum in this study. A higher score indicates a higher probability an individual is experiencing depression. A cut-off score of 16 was used to determine non-depression/depression [35]. The Chinese version of the CESD has been used with pregnant women and postpartum mothers with high Cronbach’s alphas (Cronbach’s alpha = 0.89) [36]. The Cronbach’s alpha of the CESD at T1 in this study was 0.90. 

Stress. The 10-item Perceived Stress Scale (PSS) (Appendix A) is a five-point (0–4) Likert-type scale that measures the level an individual appraises as stressful. The validity of the PSS was supported based on the result of a factor analysis that one factor could explain 77.5% of the total variance of stress [37]. The Chinese version of the PSS has been used to measure maternal stress with satisfactory reliability (Cronbach’s alpha of 0.87) [38]. The Cronbach’s alpha of the PSS at T1 in this study was 0.84.

Anxiety. The State-Trait Anxiety Inventory (STAI) developed by Spieblerger [39] was used to measure state and trait anxiety in the study. It is a 40-item, four-point (1–4) scale that measures state anxiety (20 items, STAI-S) and trait anxiety (20 items, STAI-T). The scale score ranges from 20 to 80, with a higher score indicating a higher level of trait or state anxiety experienced by an individual. The STAI has been used to measure anxiety levels of the pregnant population with satisfactory reliability [40]. A cut-off score of 40–41 was used to determine low-high level of anxiety [41]. The STAI has been translated into Chinese and its Cronbach’s alpha was high (0.90 for the STAI-S and 0.81 for the STAI-T) [42]. The Cronbach’s alpha for the STAI-S and STAI-T at T1 in this study was 0.88 and 0.93, respectively.

### 2.5. Procedure

The study was conducted after an approval from the Institutional Review Board (IRB No. 101,073 and 100–2777A3). Data were collected at the obstetric clinic in hospitals while the participants were waiting for prenatal checkups. The principal investigator explained the research to the obstetricians in the two hospitals where the research team affiliated and had their permission to recruit participants in the obstetric clinic. The obstetricians and registered nurses mentioned the study to potential participants when they were having their prenatal checkups. Those who were interested in participating in the study or had questions about the study then talked with the research investigator or trained research assistant who stayed at the waiting area of the clinic. The research, participants’ rights, confidentiality, and privacy were introduced to the participants. After agreeing and signing a consent form, participants either were interviewed or self-administered the questionnaires. Participants who preferred to complete the questionnaires at home were asked to mail the completed questionnaires to researchers using the self-addressed and stamped returning envelope within 1 week. Subsequent appointments were made after the participants had scheduled their following checkups. Participants were asked to inform the investigators when they gave birth, and another appointment for postpartum data collection was made. Phone calls were made to remind the participants of their appointments and at their due date if the participants did not inform the investigators about their labor/delivery.

### 2.6. Data Analysis

Collected data were managed and analyzed using SPSS version 23.0. The measured variables were examined for missing values and normality. The missing completely at random (MCAR), which assumes the missing value is not dependent on both observed and unobserved data, is a stronger missing data mechanism than missing at random (MAR) [43]. The Little’s MCAR test is a single test developed by Little to avoid problems of multiple comparisons when testing MCAR [44]. Results showed that the STAI-S at T1 and T2, STAI-T at T1 and T4, and CESD at T2 and T4 had one missing value. Little’s MCAR test showed that the data was “missing completely at random” (X2 = 40.29, *p* = 0.63). The expectation-maximization (EM) algorithm was used to manage missing values. The Shapiro-Wilk test showed that the PSS at T4, STAI-T at T2, and CESD at all survey times were not normally distributed. Descriptive statistics were used to understand the participants’ demographic information and levels of measured variables. Pearson and Spearman’s correlation was used to test relationships between measured variables. The Generalized Estimating Equation (GEE) was used to test changes of the measured variables by time period and prediction of stress and anxiety on postpartum depression.

## 3. Results

### 3.1. Trends of Perinatal Stress, Depressive Symptoms, and Anxiety

From Figure 1 and Table 2, stress levels decreased from T1 to T2 and T3 and then increased at T4. However, when compared with T1, the change at T2 was statistically significant. In addition, since the mean score of the PSS at T1 and T4 was extremely close, we inferred that stress at T4 was significantly higher than it was at T2. Trait anxiety increased from T1 to T4; yet, those changes were not statistically significant. State anxiety increased throughout pregnancy and decreased at postpartum. Levels of state anxiety at T3 were significantly higher than that at T1. Levels of depressive symptoms increased along the time from T1 to T4, and the level at T3 and T4 was significantly higher than that at T1.

The percentage of women experiencing depression at T1 through T4 ranged from 25.6% to 34.9%; ranged from 53.2% to 60.2% for trait anxiety; and ranged from 59.9% to 63.9% for state anxiety (Table 3). While 13.3% experienced depression at all four data collection time points, 30.1% and 34.9% of women experienced higher levels of trait anxiety and state anxiety, respectively, at all four time points. More than one fourth of women were both depressed and anxious at some point in time during pregnancy and postpartum.

### 3.2. Prediction of Stress and Anxiety on Depressive Symptoms

As shown in Table 4, all measured variables were significantly correlated with each other at all time points (r ranged from 0.35 to 0.91, *p* < 0.001 for all correlations). In order to decide which demographic variables should be included in the GEE analysis to exclude influences of demographic factors on postpartum depressive symptoms, the Mann-Whitney U test or Kruskal-Wallis test were done to compare level of depressive symptoms by demographic variables. Results showed that the level of postpartum depressive symptoms differed by parity (Mann-Whitney test = 585.50, *p* = 0.03) and sleep disturbance (Mann-Whitney test = 442.50, *p* = 0.01) (Table 5). Therefore, parity and sleep disturbance were included in the GEE analysis. As shown in Table 6, depressive symptoms, trait anxiety, and state anxiety at T1; trait anxiety at T2; and state anxiety at postpartum could predict postpartum depressive symptoms.

## 4. Discussion

This study aims to explore trends and relationships between perinatal depressive symptoms, anxiety, and stress and predictors of postpartum depressive symptoms. We found that the prevalence of depression in the women was over one fourth (ranged from 25.6% to 34.9%), and it increased over the time from pregnancy to postpartum. This result was similar to a study done in German that 27.7% of mothers self-reported that they felt depressed during pregnancy [18]; one national data analysis in the United States that 35.6% of Chinese mothers experienced depression [9]; and another study done in Taiwan [38]. However, the rate of depression was higher than that of Chinese women (19.7% during pregnancy and 14.8% in postpartum) as reported in Nisar et al.’s [16] meta-analysis article when depression were assessed using the CESD. Despite the difference in the prevalence of depression caused by different assessment tools and study regions, several reasons warrant early detection and management of prenatal depressive symptoms. First, the intensity of depressive symptoms increases over time and significantly increases at 36 weeks gestation and postpartum when the woman is approaching labor/delivery and is about to face the challenge of postpartum physical, psychosocial, and role changes as well as childcare responsibilities. Second, prenatal depressive symptoms can predict postpartum depressive symptoms. Third, depression has negative impacts on both women and their children [14,15,16,18].

Over half (ranged from 53.2% to 63.9%) of perinatal women experienced anxiety in our study, especially around 60% of women experienced state anxiety during pregnancy and postpartum. This rate was much higher than the pooled rate of perinatal anxiety symptoms (ranged from 18.2% to 24.6%) and the pooled rate of perinatal trait anxiety (ranged from 29.1% to 32.5%) reported in other studies [11]. Factors causing this high rate of perinatal anxiety in our study sample need further exploration. Regardless, in our study, the intensity of state anxiety significantly increased at T3 when the mother was about to give birth. Since prenatal anxiety is related to a higher risk of birth complications and lower Apgar scores in babies [14], and is also significantly correlated with depressive symptoms, as recommended by the American College of Obstetricians and Gynecologists (ACOG), obstetric care providers should screen pregnant women for depressive and anxiety symptoms at least once during pregnancy and postpartum. In particular, a full assessment for postpartum depression and anxiety is recommended [8].

Perceived stress was strongly correlated with anxiety and depressive symptoms during pregnancy and postpartum in our study, which was similar to the few studies that focused on perinatal perceived stress, anxiety, and depressive symptoms [26,27,38]. However, in contrast to another study’s finding that perceived stress during late pregnancy and early postpartum could predict postpartum depressive symptoms [27], we did not find this prediction in our study. The inconsistency might be due to different countries of study and measuring tools. Nevertheless, the management of perceived stress is warranted because of its strong correlation with perinatal anxiety and depressive symptoms. Other studies, which found that pregnant women with high stress levels had high odds of having probable prenatal depression and that prenatal stress could predict prenatal depressive symptoms and anxiety, [45,46] also highlighted the importance of assessing and managing perinatal stress.

The trend of perceived stress from pregnancy to postpartum found in our study was similar to another longitudinal study done in Taiwan [38] which found that stress levels decreased from 24 weeks to 32 weeks and remained at similar levels at 36 weeks of gestation, but returned to high levels after childbirth. This result was different from another study done in Australia which found that stress levels significantly decreased from 24 weeks to 28 weeks, then increased at 32 weeks and remained about the same at 36 weeks gestation [46]. These results may indicate that during the second trimester, pregnant women may feel a certain degree of relief from discomforts occurring in the first trimester and therefore decreased their stress level. However, after childbirth, new mothers are stressed since they have to face tasks of the maternal role and adapt to physical and psychosocial changes.

## 5. Limitation and Suggestion

This study uses a prospective longitudinal study design that follows pregnant women from 24 weeks gestation until one month postpartum. This design provides a more comprehensive picture of the changing courses of perinatal psychological distress. Despite the strength of the study design, there are limitations. The generalizability of the findings in our study is limited because of the relatively small sample size, high attrition (46.8% from T1 to T4), studying only in metropolitan cities, and using self-reported measurements rather than diagnostic interviews to assess mental disorders. Although post-hoc power analysis showed a high power, which showed an adequate sample size to detect effects of significance, we recommend conducting more longitudinal studies that include more pregnant women from cities in various stages of urban development. While conducting longitudinal studies, in addition to frequent contacts with the participants, reminder calls, data collected in clinic when participants arrived for checkups, and monetary incentives done in our study, it is recommended to apply strategies that show effectiveness in improving retention rates, such as using social media, using alternative methods of data collection, sending thank-you cards, and offering home visits [47].

## 6. Conclusions

The purpose of this longitudinal study is to explore trends of stress, anxiety, and depressive symptoms from pregnancy to postpartum and understand predictions of stress and anxiety on postpartum depression. We found that levels of anxiety and depressive symptoms increase from 24 weeks gestation to postpartum, whereas stress levels decrease during pregnancy but increase in postpartum. While over one fourth of women experience perinatal depressive symptoms, over half of women experience anxiety symptoms especially during late pregnancy and postpartum. More than one fourth of women were both depressed and anxious at some point of time during pregnancy and postpartum. Stress, anxiety, and depressive symptoms are significantly inter-correlated among four measuring time points. Notably, women at late pregnancy and postpartum are prone to stress, anxiety, and depression. Prenatal anxiety can predict postpartum depressive symptoms. Based on the study results, we suggest actively assessing women’s stress, anxiety, and depression beginning from early pregnancy and continuing until postpartum. We also suggest providing interventions that showed effectiveness in managing perinatal stress, anxiety, and depression to women especially at late pregnancy and postpartum period. This includes mindfulness-based childbirth and parenting programs that has significant effects in reducing perceived stress and depressive symptoms [48], internet-delivered cognitive behavioral therapy and internet-delivered behavioral activation interventions that can significantly improve perinatal depression and anxiety [49], online support, peer experiences sharing, and perinatal anxiety-related psychoeducation, which are recommended by women experiencing perinatal anxiety [50]. Care needs to be provided especially to women who are at high risk of experiencing perinatal mental distress due to certain factors: young, single or not partnered, low literacy, low socioeconomic status, having relationships or social support, experiencing prior miscarriage or perinatal loss, low self-esteem, past depression, lower income level, and higher stress [26,51].

## Figures and Tables

**Figure 1 ijerph-18-09307-f001:**
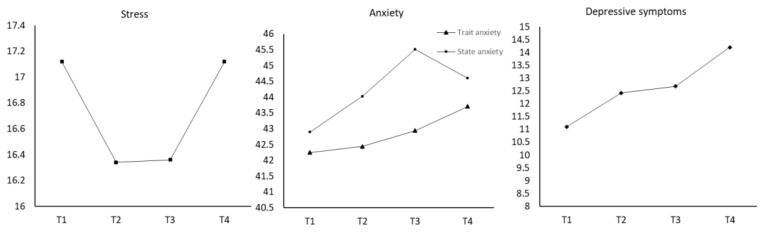
Longitudinal Changes of Perinatal Stress, Anxiety, and Depressive Symptoms.

**Table 1 ijerph-18-09307-t001:** Demographic information of the participants at Time 1 (*n* = 156).

Variable	*n*	%
Parity		
Primiparous	83	53.2
Multiparous	73	46.8
Marital status		
Single	6	3.8
Married/partnered	148	94.9
Other	2	1.3
Education level		
Bachelor’s or higher	80	51.3
Lower than bachelor’s	74	47.4
Missing	2	1.3
Employment		
Unemployed	39	25.0
Employed	117	75.0
Planned pregnancy		
No	58	37.2
Yes	95	60.9
Missing	3	1.9
Happy about the pregnancy		
Uncertain	12	7.7
Very unhappy/unhappy	7	4.5
Very happy or happy	136	87.2
Missing	1	0.6
Sleep disturbance		
No	45	28.8
Yes	110	70.5
Missing	1	0.6
Type of sleep disturbance ^1^		
Interrupted sleep	78	73.6
Unable to fall asleep	29	27.4
Lack of sleep	12	11.3
Sleepy during daytime	25	23.6

^1^ Only those who experienced sleep disturbance (*n* = 110) answered this multiple-selection question.

**Table 2 ijerph-18-09307-t002:** Trends of stress, anxiety, and depressive symptoms from pregnancy to postpartum.

Variable	M ± SD	B	SE	95% CI	Wald X^2^	*p*
Stress						
Intercept		17.12	0.42	16.30, 17.94	1676.52	<0.001
T1	17.12 ± 5.24	-				
T2	16.34 ± 5.58	−0.85	0.36	−1.56, -0.14	5.44	0.02
T3	16.36 ± 5.98	−0.79	0.41	−1.61, 0.02	3.67	0.06
T4	17.12 ± 6.52	0.25	0.62	-0.98, 1.47	0.16	0.69
Trait anxiety						
Intercept		42.25	0.76	40.67, 43.73	3113.60	<0.001
T1	42.25 ± 9.49	-				
T2	42.44 ± 10.23	0.10	0.63	−1.14, 1.35	0.03	0.87
T3	42.94 ± 9.87	0.49	0.63	-0.074, 1.73	0.61	0.43
T4	43.71 ± 10.87	1.59	1.12	-0.61, 3.78	2.01	0.16
State anxiety						
Intercept		42.91	0.88	41.19	2400.89	<0.001
T1	42.90 ± 10.97	-				
T2	44.03 ± 11.49	1.03	0.79	-0.52, 2.58	1.71	0.19
T3	45.51 ± 12.23	2.38	0.90	0.62, 4.15	7.00	0.01
T4	44.60 ± 11.89	1.94	1.23	-0.47, 4.36	2.49	0.12
Depressive symptoms						
Intercept		11.10	0.62	9.88, 12.31	319.09	<0.001
T1	11.10 ± 7.78	-				
T2	12.42 ± 9.10	1.21	0.64	-0.03, 2.45	3.65	0.06
T3	12.68 ± 8.53	1.55	0.66	0.26, 2.85	5.54	0.02
T4	14.20 ± 11.01	3.17	1.08	1.05, 5.29	8.60	0.003

T1: 23 to 28 weeks of gestation; T2: 32 to 36 weeks of gestation; T3: over 36 weeks of gestation; T4: 4 to 6 weeks postpartum.

**Table 3 ijerph-18-09307-t003:** Prevalence of anxiety and depression from pregnancy to postpartum.

Variable	T1 (*n* = 156)		T2 (*n* = 147)		T3 (*n* = 129)		T4 (*n* = 83)		All Four Times	
	*n*	%	*n*	%	*n*	%	*n*	%	*n*	%
Depression	40	25.6	42	28.6	42	32.6	29	34.9	11	13.3
Trait anxiety	83	53.2	79	53.7	76	58.9	50	60.2	25	30.1
State anxiety	94	60.3	88	59.9	82	63.6	53	63.9	29	34.9
Depression and anxiety	39	25.0	42	28.6	40	31.0	27	32.5	11	13.1

T1: 23 to 28 weeks of gestation; T2: 32 to 36 weeks of gestation; T3: over 36 weeks of gestation; T4: 4 to 6 weeks postpartum.

**Table 4 ijerph-18-09307-t004:** Correlations between measured variables.

Variable	1	2	3	4	5	6	7	8	9	10	11	12	13	14	15
T1															
1. Stress	-														
2. Trait anxiety	0.68	-													
3. State anxiety	0.66	0.84	-												
4. Depressive symptoms	0.64	0.79	0.69	-											
T2															
5. Stress ^a^	0.67	0.69	0.63	0.64	-										
6. Trait anxiety	0.60	0.72	0.69	0.63	0.82	-									
7. State anxiety ^a^	0.51	0.68	0.64	0.58	0.77	0.91	-								
8. Depressive symptoms	0.56	0.65	0.61	0.70	0.80	0.86	0.81	-							
T3															
9. Stress ^a^	0.63	0.65	0.60	0.62	0.78	0.71	0.70	0.69	-						
10. Trait anxiety ^a^	0.53	0.71	0.65	0.63	0.71	0.81	0.81	0.76	0.81	-					
11. State anxiety ^a^	0.40	0.58	0.59	0.52	0.63	0.69	0.77	0.67	0.75	0.87	-				
12. Depressive symptoms	0.48	0.62	0.52	0.62	0.65	0.70	0.67	0.79	0.78	0.83	0.76	-			
T4															
13. Stress	0.51	0.36	0.35	0.44	0.59	0.44	0.39	0.44	0.52	0.47	0.41	0.46	-		
14. Trait anxiety ^a^	0.44	0.35	0.45	0.44	0.55	0.50	0.47	0.53	0.51	0.53	0.49	0.47	0.82	-	
15. State anxiety ^a^	0.43	0.39	0.41	0.46	0.58	0.50	0.43	0.49	0.55	0.52	0.45	0.49	0.83	0.88	-
16. Depressive symptoms	0.48	0.36	0.38	0.50	0.57	0.51	0.46	0.52	0.56	0.52	0.46	0.53	0.85	0.88	0.84

Pearson correlation was performed for the variable that was normally distributed (marked with superscript letter ^a^). For variables that were not normally distributed, Spearman’s correlation was applied.

**Table 5 ijerph-18-09307-t005:** Comparison of postpartum depressive symptoms by demographic variables.

Variable	M	SD	Median	U or H	*p*
Parity				594.50	0.02
Primiparous	16.66	11.97	14.00		
Multiparous	10.82	8.59	8.00		
Marital status				1.52	0.47
Single	13.00	15.47	8.50		
Married	14.07	10.88	12.00		
others	21.50	10.61	21.50		
Education level				701.00	0.31
Bachelor or higher	12.96	10.26	11.00		
Lower than bachelor	15.85	12.10	12.00		
Employment				486.00	0.40
Unemployed	11.42	10.56	11.00		
Employed	14.65	11.15	12.34		
Planned pregnancy				658.50	0.41
No	12.32	9.66	10.00		
Yes	15.00	11.69	12.68		
Happy about the pregnancy				0.68	0.71
Very unhappy or unhappy	16.33	5.03	17.00		
Very happy or happy	14.13	11.06	12.00		
Uncertain	13.75	15.28	10.00		
Sleep disturbance				448.50	0.01
No	9.73	9.12	6.00		
Yes	16.15	11.36	14.00		

U: Mann-Whitney U test statistic for variable with two levels; H: Kruskal-Wallis H test statistic for variable with more than two levels.

**Table 6 ijerph-18-09307-t006:** Prediction of perinatal stress and anxiety on postpartum depression.

Variable	B	SE	95% CI	Wald X^2^	*p*
(Intercept)	−19.84	2.02	−23.81, −15.87	96.03	<0.001
Depression T1	−2.52	1.11	−4.70, -0.34	5.12	0.02
T2	−1.81	1.19	−4.14, 0.52	2.31	0.13
T3	−1.65	1.12	−3.85, 0.54	2.12	0.14
T4	-				
Stress T1	0.12	0.07	-0.03, 0.26	2.45	0.12
T2	0.07	0.15	−0.23, 0.36	0.19	0.66
T3	0.10	0.12	−0.14, 0.34	0.70	0.40
T4	−0.11	0.09	−0.30, 0.07	1.41	0.23
Trait anxiety T1	0.19	0.08	0.03, 0.34	5.77	0.02
T2	0.19	0.09	0.02, 0.37	4.68	0.03
T3	0.21	0.11	−0.01, 0.43	3.47	0.06
T4	0.07	0.07	−0.06, 0.20	1.15	0.29
State anxiety T1	−0.15	0.06	−0.27, −0.03	5.54	0.02
T2	−0.05	0.06	−0.17, 0.08	0.49	0.48
T3	0.04	0.06	−0.08, 0.17	0.49	0.49
T4	0.20	0.06	0.08, 0.32	10.65	0.001
Parity	−0.25	0.75	−1.72, 1.23	0.11	0.75
Sleep disturbance	1.10	0.71	−0.29, 2.50	2.41	0.12

T1: 23 to 28 weeks of gestation; T2: 32 to 36 weeks of gestation; T3: over 36 weeks of gestation; T4: 4 to 6 weeks postpartum.

## Data Availability

The data set is available online at https://data.mendeley.com/datasets/r37922phfd/1 (accessed on 2 September 2021).

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
