# Peer review of "Trends of Perinatal Stress, Anxiety, and Depression and Their Prediction on Postpartum Depression"

_ijerph, 2021, doi:10.3390/ijerph18179307_

Round 1

Reviewer 1 Report

This study explores trends of three main issues (stress, anxiety, and depressive symptoms) of postpartum women. It is longitudinal study

T1: 23-28 weeks gestation

T2: 32-36 weeks 14 gestation

T3: over 36 weeks gestation

T4: postpartum (T4)

One-hundred-fifty-six women were involved, and the authors were applied T-test, one-way ANOVA, Mann-Whitney test, or Kruskal-Wallis.

Although the idea could be promising, the paper poses too many sever flaws:

  1. In line 22, the author mentioned “Stress, anxiety, and depressive symptoms were inter-correlated”. How did they find it based on their analysis?
  2. Data were collected from whole of China? When did they collect their data? Please explain more about sampling and how did they approach the participants?
  3. In line 200, the authors mentioned they applied one-way ANOVA. Please provide the outputs of this analysis in a Table which is highlights values of SS, MS, F-Test, and etc.
  4. It’s better how did they find this T1, T2, T3, and T4 this categorical time and why?
  5. Please explain about MCAR test.
  6. In Section the author explained how they measured Depressive symptoms, Stress, and Anxiety. However, they haven’t mentioned how they measured postpartum depression.
  7. It’s better to provide questionnaire as a supplementary file (English version).
  8. It’s better to provide limitation of the study and suggestions for future studies.

Minor

Total percentage of “Marriage status” in Table 1 is not equal 100%.

Author Response

Response to Reviewer 1 Comments

Thank you for reviewing our manuscript, providing comments and suggestions, and willingness to allow us to revise and resubmit the manuscript. Please see below for our responses to your comments. We hope that we have addressed your major concerns.

Point 1: In line 22, the author mentioned “Stress, anxiety, and depressive symptoms were inter-correlated”. How did they find it based on their analysis?

Response 1: The correlations between measured variables were tested using Pearson or Spearman’s correlation based on whether the variable was normally or not normally distributed. To clarify how measured variables were correlated, we have added the correlation table as Table 4 in the text (page 7 and 8, Lines 237-239-237).

Point 2: Data were collected from whole of China? When did they collect their data? Please explain more about sampling and how did they approach the participants?

Response 2: The data were collected in two metropolitan cities in northern (Taipei) and southern (Chiayi) Taiwan. We have added the city names in the Setting section (page 3, line 108).

Data were collected from May 2012 to September 2013. We have added this information in the Design section (page 3, lines 93-94).

Non-probability sampling was used to recruit participants. The principal investigator explained the research to the obstetricians in the two hospitals where the research team affiliated and had their permission to recruit participants in the obstetric clinic. The obstetricians and registered nurses mentioned the study to potential participants when they were having their prenatal checkups. Those who were interested in participating in the study or had questions about the study then talked with the research investigator or trained research assistant who stayed at the waiting area of the clinic. The research, participants’ rights, confidentiality, and privacy were introduced to the participants. Data were collected after the participants signed the consent form. We have added this information in the Setting and Procedure sections (page 3, line 110 and page 5, lines 169-176).

Point 3: In line 200, the authors mentioned they applied one-way ANOVA. Please provide the outputs of this analysis in a Table which is highlights values of SS, MS, F-Test, and etc.

Response 3: We are sorry for not presenting the comparison table. We have added the comparison table as Table 5 in the text (page 8, line 240-242). Because the variable postpartum depressive symptoms was not normally distributed, we used either the Mann-Whitney U test or Kruskal-Wallis test to compare its difference by demographic variables (page 7, lines 229-231). We reported M, SD, Median, U or H, and p and corrected the error of using t-test, one-way ANOVA.

Point 4: It’s better how did they find this T1, T2, T3, and T4 this categorical time and why?

Response 4: Pregnant women may experience more physical symptoms during the third trimester when compared with the second trimester. In addition, based on literature, the prevalence of depression remained high from the second to the third trimester whereas the prevalence of anxiety increased in the third trimester. Therefore, the study was designed to survey one time in the second trimester (before 28 weeks gestation) and two times (one month apart after 28 weeks gestation) in the third trimester. However, the survey time varied among the participants because of their availability for survey appointments. The time for the postpartum survey was done after the participants were discharged from the hospital when they had to face the childcare responsibility and adjust to the new maternal role. Since new mothers would have their postnatal checkup at around 4 to 6 weeks postpartum, the survey was set at that time point. We have added this information in the Design section (page 3, lines 96-105).

Point 5: Please explain about MCAR test.

Response 5: The missing completely at random (MCAR), which assumes the missing value is not dependent on both observed and unobserved data, is a stronger missing data mechanism than missing at random (MAR) (Li, 2013). The Little’s MCAR test is a single test developed by Little to avoid problems of multiple comparisons when testing MCAR (Little, 1988) and can be performed in SPSS. We have added this information in the Data analysis section (page 5, lines 188-192).

Point 6: In Section the author explained how they measured Depressive symptoms, Stress, and Anxiety. However, they haven’t mentioned how they measured postpartum depression.

Response 6: Postpartum depression was measured using the CESD, which was used to measure prenatal depressive symptoms. By using the same instrument, a comparison between different data collection time points could be done. A cut-off score of 16 for depression was used to categorize non-depression/depression based on the scale development study. We have added this information in the Instrument section (page 4, lines 142-145).

Point 7: It’s better to provide questionnaire as a supplementary file (English version).

Response 7: As suggested, we have provided an English version of the questionnaires as supplementary files.

Point 8: It’s better to provide limitation of the study and suggestions for future studies.

Response 8: The strength and limitations of the study and suggestions for future studies are presented in the last paragraph of the Discussion section. To make it clearer, we have added a subtitle as Limitation and Suggestion to the paragraph (page 10, lines 300-315-313).

Point 9: Total percentage of “Marriage status” in Table 1 is not equal 100%.

Response 9: We appreciate the reviewer for pointing out this error. We have corrected the percentage for “others” (page 3, line 137). 

Reviewer 2 Report

It has been my pleasure to review this manuscript on perinatal stress, anxiety and depression.

Unfortunately, mental health during pregnancy is a subject very little studied and it is not given the importance it deserves. The pregnant woman must be studied comprehensively and not only from the obstetric point of view.

In general, the manuscript is well developed.
The introduction seems correct to me. It puts us in the context of the situation. At the end of the introduction, I would suggest that the purpose of this research be stated. This objective is defined at the beginning of the discussion section. I think it would be more interesting to express it in the introduction.

In the material and method section, the design seems correct to me, the selection of the sample is well explained and the instruments used to measure depressive symptoms and stress seem to have good psychometric properties.

It would have been interesting to explain who developed these measurement instruments for the first time and then explain who and when it was validated in the language of the participants.

The statistical study seems correct to me. The results are clearly stated, with regression analysis being undoubtedly the strongest point for predicting stress and anxiety trends in postpartum depression.

The discussion seems adequate to me, recognizing at the end of this section the limitations found.

The conclusions also seem correct to me, correctly supporting the results obtained.

Thanks

Kind Regards

Author Response

Response to Reviewer 2 Comments

Thank you for reviewing our manuscript, providing comments and suggestions, and willingness to allow us to revise and resubmit the manuscript. Please see below for our responses to your comments. We hope that we have addressed your major concerns.

Point 1: Unfortunately, mental health during pregnancy is a subject very little studied and it is not given the importance it deserves. The pregnant woman must be studied comprehensively and not only from the obstetric point of view.

In general, the manuscript is well developed.

The introduction seems correct to me. It puts us in the context of the situation. At the end of the introduction, I would suggest that the purpose of this research be stated. This objective is defined at the beginning of the discussion section. I think it would be more interesting to express it in the introduction.

Response 1: We appreciate the reviewer’s comments and suggestions. As suggested, we have added the purpose of the research in the Introduction section (page 2, line 84-86).

Point 2: In the material and method section, the design seems correct to me, the selection of the sample is well explained and the instruments used to measure depressive symptoms and stress seem to have good psychometric properties.

It would have been interesting to explain who developed these measurement instruments for the first time and then explain who and when it was validated in the language of the participants.

Response 2: We appreciate the reviewer’s pointing out the missing information about the Chinese version of all measurements. We have added the information in the Instrument section (page 4, line 145 and 152; page 5, line 161-162).

Point 3: The statistical study seems correct to me. The results are clearly stated, with regression analysis being undoubtedly the strongest point for predicting stress and anxiety trends in postpartum depression.

The discussion seems adequate to me, recognizing at the end of this section the limitations found.

The conclusions also seem correct to me, correctly supporting the results obtained.

Response 3: We appreciate the reviewer’s comments.

Round 2

Reviewer 1 Report

The authors had adequately revise the manuscript point by point